

# Aboveground insect herbivory increases plant competitive asymmetry, while belowground herbivory mitigates the effect

Pernilla Borgström, Joachim Strengbom, Maria Viketoft and Riccardo Bommarco

Department of Ecology, Swedish University of Agricultural Sciences, Uppsala, Sweden

## ABSTRACT

Insect herbivores can shift the composition of a plant community, but the mechanism underlying such shifts remains largely unexplored. A possibility is that insects alter the competitive symmetry between plant species. The effect of herbivory on competition likely depends on whether the plants are subjected to aboveground or belowground herbivory or both, and also depends on soil nitrogen levels. It is unclear how these biotic and abiotic factors interactively affect competition. In a greenhouse experiment, we measured competition between two coexisting grass species that respond differently to nitrogen deposition: *Dactylis glomerata* L., which is competitively favoured by nitrogen addition, and *Festuca rubra* L., which is competitively favoured on nitrogen-poor soils. We predicted: (1) that aboveground herbivory would reduce competitive asymmetry at high soil nitrogen by reducing the competitive advantage of *D. glomerata*; and (2), that belowground herbivory would relax competition at low soil nitrogen, by reducing the competitive advantage of *F. rubra*. Aboveground herbivory caused a 46% decrease in the competitive ability of *F. rubra*, and a 23% increase in that of *D. glomerata*, thus increasing competitive asymmetry, independently of soil nitrogen level. Belowground herbivory did not affect competitive symmetry, but the combined influence of above- and belowground herbivory was weaker than predicted from their individual effects. Belowground herbivory thus mitigated the increased competitive asymmetry caused by aboveground herbivory. *D. glomerata* remained competitively dominant after the cessation of aboveground herbivory, showing that the influence of herbivory continued beyond the feeding period. We showed that insect herbivory can strongly influence plant competitive interactions. In our experimental plant community, aboveground insect herbivory increased the risk of competitive exclusion of *F. rubra*. Belowground herbivory appeared to mitigate the influence of aboveground herbivory, and this mechanism may play a role for plant species coexistence.

## INTRODUCTION

Herbivory and soil nutrient availability are two factors that strongly influence competitive relationships between plant species (*Keddy, 2007*). Insects are important consumers of plant biomass in essentially all terrestrial ecosystems, and there is ample evidence of

Corresponding author
Pernilla Borgström,
pernilla.borgstrom@slu.se

major impacts of insect herbivory on single plant populations (*Maron & Crone, 2006*). Additionally, there is evidence from field experiments demonstrating shifts in community composition under insect herbivory (*Brown & Gange, 1989*; *Carson & Root, 2000*; *Van Ruijven et al., 2005*; *Allan & Crawley, 2011*). Theory suggests that such community-level shifts are driven by changes in interspecific competition among plants, as even low rates of biomass removal by insects can alter the competitive hierarchy of a plant community (*Louda, Keeler & Holt, 1990*). For example, insect herbivory might shift plant community composition by facilitating competitive exclusion of one plant species by another (*Kim, Underwood & Inouye, 2013*), but it can also reduce the competitive potential of a dominant species in a plant community, thereby facilitating coexistence and enhancing species diversity (*Carson & Root, 2000*). There is, however, little experimental evidence specifically detailing to what degree, and under which environmental circumstances, insect herbivory mediates plant competition (*Sotomayor & Lortie, 2015*).

A change in soil nutrient levels caused, for example, by atmospheric nitrogen deposition, can have pervasive impacts on plant community composition (*Stevens et al., 2004*; *Bobbink et al., 2010*; *Cleland & Harpole, 2010*). An important underlying mechanism is probably that increased nitrogen availability causes shifts in aboveground competitive interactions among plants (*Vallano, Selmants & Zavaleta, 2012*), as competition for light increases (*Hautier, Niklaus & Hector, 2009*). Changes in nitrogen levels can also affect herbivory, and the impact of herbivory on plant competition. For example, increased nitrogen availability can lead to increased consumption rates and increased population growth of herbivores (*Throop et al., 2004*). Herbivores have the potential to prevent competitive exclusion of species that are competitively disfavoured by increased nutrient availability (*Ghedini, Russell & Connell, 2015*), and to counteract the intensified competition for light that nutrient addition causes (*Borer et al., 2014*). However, there is very little understanding of how changes in nutrient availability together with insect herbivory above and below ground determine the outcome of interspecific plant competition.

Herbivorous insects exert pressure on plants both above and below ground, a fact most often disregarded as each subsystem has predominantly been studied in isolation, with a bias towards aboveground herbivory (*Blossey & Hunt-Joshi, 2003*). Competition between plants aboveground is thought to be size-asymmetric, meaning that larger, fast-growing individuals have a disproportionately large competitive advantage over smaller, slow-growing individuals (*Weiner, 1990*; *Schwinning & Weiner, 1998*). A second definition of competitive asymmetry (*Keddy & Shipley, 1989*) is unrelated to size, and occurs when one species is a stronger interspecific than intraspecific competitor. Since herbivores often preferentially feed on fast-growing, highly competitive species (*Huntly, 1991*), aboveground herbivory can be expected to reduce competitive asymmetry (irrespective of definition) between plant species. One consequence of this could be delayed competitive exclusion of slower growing species, especially under conditions of high soil nitrogen, when aboveground competition is likely to be at its strongest (*Hautier, Niklaus & Hector, 2009*), i.e., when competitive asymmetry is high. This, however, remains to be tested.

The effect of belowground herbivory on plant competition has received little attention, although there is evidence of dominance shifts between competing plant species when

they are subjected to root herbivory in the field (*Roubíčková, Mudrák & Frouz, 2012*). Root competition for soil nutrients is thought to be size-symmetric, meaning that the competitive advantage of a plant species below ground is proportional to the size of the plant (*Casper & Jackson, 1997*; *Cahill & Casper, 2000*). A generalist root herbivore may feed more on the roots of the plant species that is competitively dominant in the soil, simply because it will encounter the roots of that species more often than the roots of subdominant species. We propose that belowground insect herbivory therefore has the greatest influence on plant competition at low nutrient levels, at which belowground competition for resources is believed to play a greater role for the competitive outcome (*Cahill, 1999*).

Aboveground herbivory can either reduce (*Bardgett, Wardle & Yeates, 1998*; *Blue et al., 2011*) or increase (*Pucheta et al., 2004*) plant growth below ground. Belowground herbivory most often reduces plant growth aboveground (*Blossey & Hunt-Joshi, 2003*; *Ladygina et al., 2010*; *Tsunoda, Kachi & Suzuki, 2014*). Studies on plant population-level effects of simultaneous above- and belowground herbivory have documented additive effects on plant overall fitness (*Maron, 1998*) and biomass production (*He, Ding & Lu, 2014*), and non-additive effects on reproductive output (*Poveda et al., 2003*). In a plant community, simultaneous herbivory from insects above- and belowground can influence species diversity in a way not predicted by the individual effect of each herbivore (*Van Ruijven et al., 2005*), but such interactive effects are not necessarily evident on competition between the species in the plant community (*Jing et al., 2015*), and the combined effect of above- and belowground herbivory on interactions between plants remains largely unexplored.

We investigated how above- and belowground insect herbivory influenced shoot competition between two grass species. We focused on shoot competition, as it is known to be central in determining a plant community's response to nitrogen deposition (*Hautier, Niklaus & Hector, 2009*). We used the fast-growing *Dactylis glomerata* L., which is favoured by nutrient-rich soils, and the slow-growing *Festuca rubra* L., favoured by nutrient-poor soils (*Van der Werf et al., 1993*; *Duru, Cruz & Magda, 2004*). These species commonly coexist in European grasslands. In nature, *D. glomerata* is expected to outcompete *F. rubra* when soil nitrogen availability increases. We examined how the competitive interaction between these two plant species changed due to above- and belowground insect herbivory at two contrasting soil nitrogen levels. We tested the prediction that aboveground herbivory would counteract an increased competitive advantage of *D. glomerata* brought about by simulated nitrogen deposition, i.e., it would decrease aboveground competitive asymmetry (*sensu Keddy & Shipley, 1989*). We predicted that belowground herbivory, conversely, would counteract the competitive advantage of *F. rubra* at low soil nitrogen levels, which would be reflected by the aboveground competition response. We expected the effect of aboveground herbivory to remain after cessation of herbivore feeding, as the herbivores would exert a lasting effect on plant competition by decreasing aboveground competitive asymmetry.

## METHODS

### Plant species

Seeds of *Festuca rubra* and *Dactylis glomerata* were obtained from a commercial seed producer (Herbiseed, Twyford, United Kingdom) and sown in planting trays, ca 20 seeds

per cavity, in May 2014. *Dactylis glomerata* was sown one day earlier due to having a slower germination time than *F. rubra*. After 3 weeks, the species were replanted into 1 L plastic pots ($\phi$13.2 cm, depth 10.7 cm) filled with a 1:1 mixture of standard potting soil and sand. The nitrogen content of the soil mixture was ca 0.06%, with a C:N ratio of 37.4. The plants were kept in a greenhouse with a 16 h light cycle, at 18 °C.

## Experimental design

To measure how competition was affected by insect herbivory and nitrogen, we used a replacement design (*De Wit, 1960*), including monocultures of each species, and 0.5:0.5 mixtures. Each pot consisted of four small stands of grass. A monoculture pot contained four stands of one species, while a mixture pot contained two stands of each species in a mixture. A stand consisted of 5–32 tillers. The stands within a pot were matched according to size (by visual approximation) at the time of planting, to ensure that any differences in competitive outcome were not a result of differences in initial size. One experimental unit consisted of a triplet of one mixture pot, and a pot each of the two monocultures.

With a replacement design, it is possible to assess how growth is affected when a species is grown with conspecific neighbours compared with when it is grown together with another species (*Cousens, 1996*). The three treatments (aboveground herbivory, belowground herbivory, and nitrogen) were crossed, in a fully factorial design, which also included an untreated control. Each experimental unit (i.e., each triplet) was assigned one of the eight treatment combinations. Each treatment combination was replicated eight times, giving a total of 64 replicates.

## Treatments

For the aboveground herbivore treatment, we used the larval stage of a noctuid moth, *Spodoptera littoralis,* Boisduval. This species is a generalist herbivore native to Africa, the Middle East and Southern Europe, and known to feed on more than 44 plant families (*Ellis, 2004*). It has previously been used, for example, in assessments of the competition-defense trade-off among plants (*Kempel et al., 2011*). The duration of the larval stage is strongly dependent on temperature and climate, but lasts approximately four weeks at 18 °C (Anderson, pers. comm., 2015). Specimens were obtained as newly-hatched larvae from a breeding culture in the lab, and kept on a diet of lettuce for 9 days before the start of the experiment. The larvae were in instar 4–5 at the start of the experiment and were randomly allocated to experimental pots to ensure an even spread of larval sizes. The aboveground herbivory treatment consisted of two *S. littoralis* individuals per pot. Due to the fact that *S. littoralis* often consumed all or most of an entire strand of a species at a time, it was not possible to reliably estimate feeding damage.

For the belowground herbivore treatment we used wireworms (larvae of *Agriotes* spp., L; Coleoptera: Elateridae). Wireworms are common generalist root herbivores in European grasslands. They have been experimentally shown to affect the feeding behaviour of *S. littoralis*, when the two spatially separated herbivores feed on the same plant (*Anderson, Sadek & Wackers, 2011*). Specimens were obtained from a breeding culture (Applied Plant Research, PPO, Wageningen UR, Wageningen, Netherlands), and kept refrigerated in

soil until the start of the experiment. The belowground herbivore treatment consisted of two *Agriotes* spp. individuals per pot. Specimens were not all of the same life stage and were distributed randomly over experimental pots, with an effort to obtain an even size distribution across treatments.

Nitrogen was added in the form of ammonium nitrate solution, corresponding to a simulated N deposition of ca 17 kg/ha. This deposition level lies in the middle of the range of N deposition to which West European grasslands are currently exposed. (e.g., *Stevens et al., 2010*). The nitrogen treatment was added 2.5 weeks after the plants were transferred to their experimental pots; the herbivory treatments began one day after that.

## Harvest

Pots were harvested a first time three weeks after herbivore addition, when *S. littoralis* was nearing pupation. The plants were cut at 2 cm height above the soil surface. The material was dried at 65 °C for 48 h and weighed. At the first harvest, all *S. littoralis* individuals were removed from the pots that were assigned to an aboveground herbivory treatment. Belowground herbivores (along with a few *S. littoralis* individuals that had gone into pupation in the soil) were left in the pots, as they were impossible to extract from the soil without damaging the root systems of the plants. We opted against using soil insecticides to remove the belowground herbivores, as we were wary of the potential, indirect fertilizing effects that insecticide application can have, when the targeted organisms die and release nutrients into the soil (*Swift & Anderson, 1993*).

After the first harvest, plants were allowed to regrow without aboveground herbivores, and were cut, dried and weighed after a further three and six weeks.

## Measure of competition

To measure competition intensity, we used the aggressivity index (*McGilchrist & Trenbath, 1971*). The aggressivity index uses the relative yield (henceforth RY) of each species to give a measure of the strength and direction of a competitive interaction between two plant species. RY is calculated by dividing the yield (Y) of a species grown in mixture with that of the same species grown in monoculture. Thus, the respective RY values are an indicator of competitive asymmetry, *sensu* *Keddy & Shipley (1989)*. Eq. (1) shows the calculation of RY for a hypothetical species *a*, grown with and without a competing species *b*. Aggressivity (Eq. (2)) is then calculated by subtracting the RY of species *a* with that of its competitor, species *b*. Further information about the calculation of RY and aggressivity can be found in Fig. S1.

$$\text{RY}_a = Y_{ab}/Y_{aa} \tag{1}$$
$$\text{Aggressivity} = \text{RY}_a - \text{RY}_b \tag{2}$$

## Statistical analysis

To test effects of herbivory and nitrogen on aggressivity at the first harvest, we applied a linear mixed effects model to the data, with nitrogen, aboveground herbivory and belowground herbivory as fixed effects, and block as a random effect. We started by

applying the full model, including second- and third grade interactions between the three variables, and then simplified the model by eliminating non-significant parameters (significance level $\alpha = 0.05$). We corrected for heteroscedasticity of residuals by applying a variance structure that used aboveground herbivory as a grouping factor. To understand where the effects of herbivory on aggressivity were originating, we analysed the RY values of each species using the same test outlined above.

To test whether aboveground herbivory had effects on competition that persisted after herbivore removal, we used a repeated measures mixed model. In the fixed-effects part of the model we included nitrogen, herbivory and harvest number. Herbivory was treated as a single factor with four levels (i.e., aboveground, belowground, above- and belowground, and 'no herbivory'), to allow for post-hoc comparisons between groups. Harvest time was treated as a numeric variable. Block was included as a random factor, and the i.d. for each triplet of mix- and monoculture pots was nested within block. Since we were interested in changes in herbivory effects over time, and at different nitrogen levels, we included all possible second-grade interactions between variables, and then simplified the model through elimination of non-significant parameters. To account for the temporally correlated structure of the data, where the aggressivity values for each subject were correlated in time, we added a general correlation structure to the model. The *glth* function from the 'multcomp' package (*Hothorn, Bretz & Westfall, 2008*) in R was used to compare the 'no herbivory' control with the aboveground herbivory treatment, and the belowground herbivory treatment with the combined above- and belowground herbivory treatment, in a Tukey's post-hoc test. We then tested for herbivory effects on the RY values for *D. glomerata* and *F. rubra* respectively, using a repeated measures mixed model. The most appropriate correlation structure was chosen for each species model based on comparison of AIC values.

To test if our belowground herbivory treatment had an effect on plant growth above ground, we inspected the monocultures of each species for differences in aboveground biomass production between the 'no herbivory' controls and belowground herbivory-treated pots. To test if belowground herbivory had an effect first at later harvest occasions we used a repeated measures mixed model.

All data analysis was performed using the 'nlme' package (*Pinheiro et al., 2015*) in R version 3.1.0 (*R Core Team, 2014*).

## RESULTS

The competitive interaction between *D. glomerata* and *F. rubra*, as measured by the aggressivity index, was altered by herbivory (Fig. 1A and Table 1), such that *D. glomerata* benefited from the presence of herbivores. Above- and belowground herbivory both had an effect on competition (Table 1). Additionally, there was a non-additive effect of above- and belowground herbivory on aggressivity when applied together, such that the effect of combined above- and belowground herbivory was smaller than would be expected from the individual effects of each herbivory treatment (Fig. 1A and Table 1). Nitrogen addition did not alter competition, either on its own or in interaction with herbivory, in

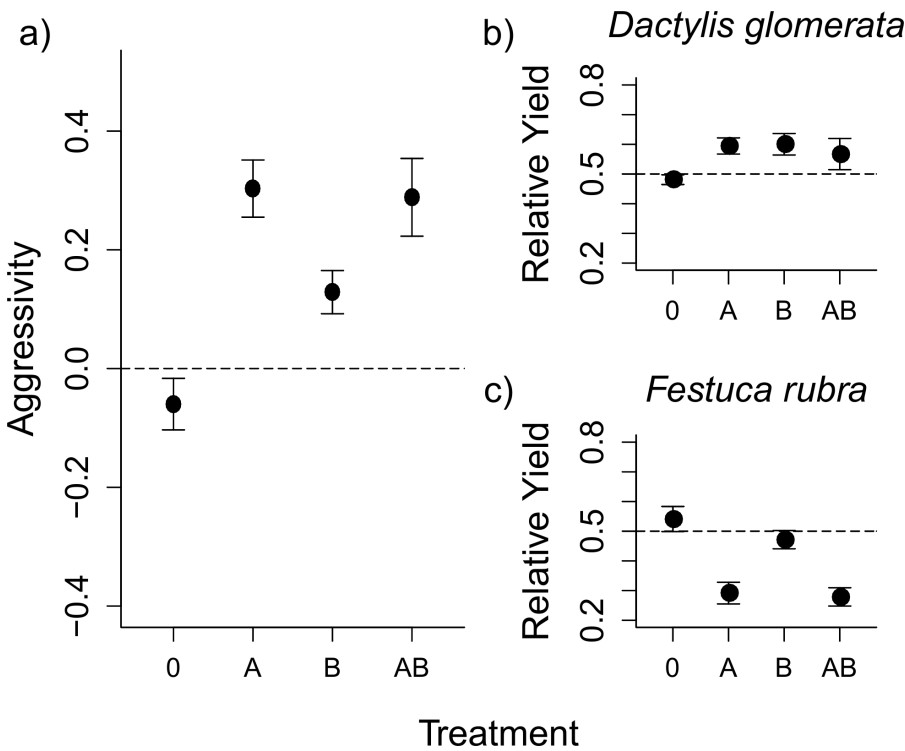

**Figure 1** **Aboveground herbivory increased competitive asymmetry, mainly by reducing the competitive ability of *F. rubra*.** The graph shows mean values ± s.e. for the competition index 'aggressivity' under different herbivory treatments (0, control; A, aboveground herbivory only; B, belowground herbivory only; AB, above- and belowground herbivory), and the Relative Yield (RY) values for each species. Aggressivity values are derived by subtracting RY values of *F. rubra* from RY values of *D. glomerata*. The positive aggressivity values under herbivory in (A) imply a change in competitive outcome that favours *D. glomerata*. The dashed line shows Aggressivity = 0, i.e., "no competition". The RY values for each species in (B) and (C) show that the shift in competition detailed in (A) is a result of decreases in RY for *F. rubra*, rather than increases in RY for *D. glomerata*. The dashed lines here show RY = 0.5, i.e., the value indicating no effect of competition on the species.

**Table 1** **Analyses of the main and interactive effects of the three treatments on aggressivity and relative yields of *Dactylis glomerata* and *Festuca rubra*.**

| Treatment | Aggressivity | | Relative yield | | | |
|---|---|---|---|---|---|---|
| | | | *Dactylis glomerata* | | *Festuca rubra* | |
| | *F*-value | *P*-value | *F*-value | *P*-value | *F*-value | *P*-value |
| **N** | | | 2.83 | 0.099 | | |
| **A** | 27.87 | <0.0001 | 1.32 | 0.26 | 39.00 | <0.0001 |
| **B** | 6.92 | 0.011 | 5.06 | 0.029 | | |
| **A × B** | 4.21 | 0.045 | 4.37 | 0.041 | | |

**Notes.**
N, Nitrogen; A, Aboveground herbivory; B, Belowground herbivory.

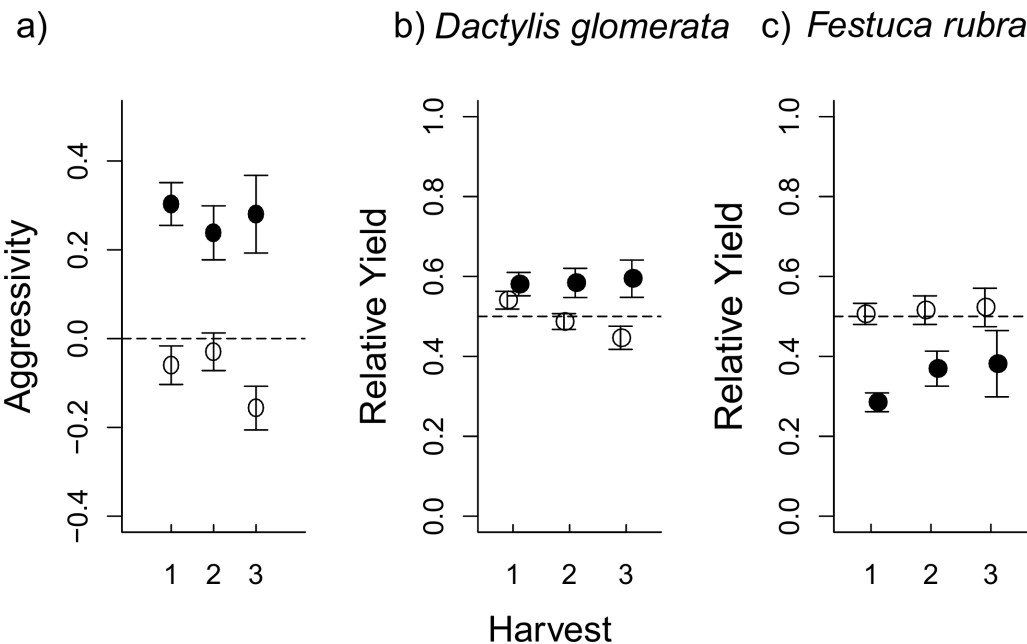

a)        b) *Dactylis glomerata*   c) *Festuca rubra*

Harvest

**Figure 2** **The effect of aboveground (AG) herbivory on competition was still seen after removal of AG herbivores at the first harvest.** In (A), aggressivity under aboveground herbivory (filled circles) is higher than in herbivore-free controls (open circles) at all three harvests, i.e., *D. glomerata* is favoured by above-ground herbivory even after herbivores are removed. This effect is a result of RY values for *D. glomerata* (B) remaining steadily higher under herbivory compared to control pots, where RY actually decreases at each consecutive harvest. Conversely, in *F. rubra* (C), RY values in herbivory pots were low at the first har-vest, before increasing at subsequent harvests following herbivore removal; while RY in control pots stays steady around the 0.5 line (i.e., "no competitive effect"). The plots show mean values $\pm$ s.e.

this or subsequent models. The nitrogen treatment was therefore dropped from further analyses, and data presented constitutes pooled values for the two nitrogen treatments, with an exception in the analysis of relative yield for *D. glomerata*. The results from the full models, including nitrogen, are presented in Tables S3–S4 for comparison. The raw mean values of aboveground biomass for each species in monoculture and in mixture under each treatment are presented in Table S5.

The changes in aggressivity under aboveground herbivory at the first harvest were a result of a reduction in the relative yield (RY) of *F. rubra*, rather than an increase in RY of *D. glomerata* (Figs. 1B–1C and Table 1). Aboveground and belowground herbivory had an interactive effect on RY of *D. glomerata* (Table 1). Aboveground herbivory influenced competition even after removal of the *S. littoralis* larvae (Fig. 2 and Table 2). The overall effect of herbivory remained the same over the three harvests, and there was no interaction between herbivory and harvest time, demonstrating that short-term exposure to herbivores generated a longer-term effect on competition. A Tukey's post-hoc comparison showed that the 'no herbivory' control and aboveground herbivory treatments differed over the three time steps, while belowground herbivory and the combined above- and belowground herbivory treatments did not (Table 2).

The consistent difference in aggressivity values (Fig. 2A) were a result of contrasting changes in RY for each species over the three consecutive harvests. In *D. glomerata*,

**Table 2** Analyses of the main and interactive effects of herbivory and harvest on aggressivity and relative yields of *Dactylis glomerata* and *Festuca rubra* at the three harvests.

| Treatment | Aggressivity | | Treatment | Relative yield | | | |
| --- | --- | --- | --- | --- | --- | --- | --- |
| | | | | *Dactylis glomerata* | | *Festuca rubra* | |
| | *F*-value | *P*-value | | *F*-value | *P*-value | *F*-value | *P*-value |
| **Herb**[a] | 13.68 | <0.0001 | **A** | | | 10.76 | 0.0018 |
| *0-A* | | 0.001 | **B** | 2.56 | 0.12 | | |
| *B-AB* | | 0.29 | **Harv** | 3.15 | 0.078 | 2.32 | 0.13 |
| **Harv**[b] | 4.89 | 0.029 | **A × Harv** | 10.24 | 0.0017 | | |

Notes.

A, Aboveground herbivory; B, Belowground herbivory..
[a] Herbivory treatment with levels 0 (control), A, B, and AB.
[b] Harvest time with levels 1–3.

there was an interaction between aboveground herbivory and harvest (Table 2), such that RY remained stable in aboveground herbivory-treated pots even after aboveground herbivore removal, while RY in control pots successively decreased (Table 2 and Fig. 2B). Aboveground herbivory caused a continuous decrease in RY of *F. rubra* even after aboveground herbivore removal (Table 2 and Fig. 2C). There was no effect of belowground herbivory on RY of either species at any of the three harvests (Fig. S2).

## DISCUSSION

We found a strong influence of insect herbivory on the competitive symmetry between two plant species, an effect which persisted well after the herbivores were removed. Aboveground herbivory increased the relative competitive ability of *D. glomerata* by reducing the competitive ability of *F. rubra*, as the latter was greatly preferred by the aboveground herbivores when the two species were grown in mixture. The effect of aboveground herbivory was, contrary to our predictions, independent of soil nitrogen level. Moreover, the combined effect of above- and belowground herbivory on competition was non-additive, as it was weaker than their individual effects would predict.

The observed increased competitive asymmetry, caused by aboveground herbivory, contradicted our prediction that herbivory should reduce competitive asymmetry between plant species, as herbivores predominantly fed on the slower-growing, less competitive *F. rubra*. These results are in accordance with both empirical (*Bentley & Whittaker, 1979*) and theoretical (*Kim, Underwood & Inouye, 2013*) evidence showing that preferential feeding of herbivores can increase the competitive asymmetry between plant species. However, it contradicts results from a biological weed control experiment, where insect herbivores instead reduced competitive asymmetry between species (*Van, Wheeler & Center, 1998*). These diverging results may be explained by differences among the initial relative competitive abilities of the studied plant species. In the biological control experiment, competitive asymmetry between the two focal species was high when herbivores were not present, but the preferential feeding of the herbivores on the dominant plant species balanced the competitive interaction between the two species (*Van, Wheeler & Center,*

*1998*). However, in our experiment the competitive interaction between *D. glomerata* and *F. rubra* remained relatively constant when herbivores were absent, with *F. rubra* having a slight competitive advantage (see Fig. 1). The strong, general preference of the aboveground herbivore for *F. rubra* drastically reduced its competitive ability, thus increasing competitive asymmetry between the two species. In our experiment, it was impossible to reliably quantify feeding damage on the grasses, and we are therefore unable to fully disentangle the effect of changing competitive interactions from those of differential consumption under different contexts. However, we were ultimately interested in assessing the competitive outcome between the two plant species. The conclusions regarding competitive outcome in our experiment would be the same, irrespective of possible information about potential diet shifts of the herbivores under the different treatments.

Belowground herbivory increased aggressivity to a lesser degree than aboveground herbivory, but in the combined treatment it appeared to reduce the impact of aboveground herbivory on competitive asymmetry. Since the relative importance of root competition in determining competitive outcome often increases with time (*Bastow Wilson, 1988*), it is possible that this balancing effect of belowground herbivory also increases with time, thus contributing to the maintenance of high species diversity in plant communities. Interactive effects between above- and belowground herbivory have previously been found to affect plant community composition, since different plant species respond differently to simultaneous application of the two herbivory types (*Van Ruijven et al., 2005*). Our study confirms such non-additive effects of above- and belowground herbivory, and demonstrates that interactive effects between the two herbivory types can also act at a finer scale, i.e., by influencing competition between two functionally similar plant species. The explanation for the interaction between the two herbivory types in our experiment might be that the above- and belowground herbivores had contrasting feeding preferences, and thus exerted counteracting effects on competition. The aboveground herbivores fed predominantly on *F. rubra*. We could not monitor the feeding of the belowground herbivores, but results from other studies show that *F. rubra* is not preferred if there are other alternatives available (*Hemerik, Gort & Brussaard, 2003*; *Roubíčková, Mudrák & Frouz, 2012*), and the wireworms in our experiment may have preferred *D. glomerata*. In a contrasting scenario, where above- and belowground herbivores exhibit the same preference, their combined effects may instead additively decrease the competitive ability of the preferred species (*Knochel, Monson & Seastedt, 2010*).

Aboveground herbivory continued to exert an effect on competition after removal of the herbivores. This supports our prediction that changed competitive asymmetry caused by herbivory is likely to remain even after the herbivores have stopped feeding. Most insect herbivores in grasslands do not feed over the entire season, and thus exert direct effects on interactions between plants only for a brief period. This may be why insect herbivory has been considered by some to have little effect on plant community composition. We show that early suppression of a plant species by herbivory can lead to a persistent effect on its competitive ability relative to other species. Even low levels of herbivory under brief, but crucial, periods of plant growth might be able to drive plant community dynamics, and may therefore ultimately play a role in the coexistence of plant species. However, to

obtain more accurate predictions on the long-lasting effects of herbivory on competition would require experiments that are conducted over longer time scales than ours, as well as modelling approaches to predict coexistence between plant species in the longer term (e.g., *Kim, Underwood & Inouye, 2013*).

Our experiment included consecutive harvests of the same experimental pots. The reason for the repeated harvests was to assess whether or not aboveground herbivory continued to affect competitive asymmetry, even after herbivore removal. Many insect herbivores feed only during a short period of the growing season, and if exposure to herbivory is brief, the overall effect of that herbivory on the plant community might be limited. However, due to the proposed asymmetrical properties of aboveground competition (*Weiner, 1990*; *Schwinning & Weiner, 1998*), an early competitive advantage aboveground, for example at the beginning of the season, is likely to remain, or amplify, as the growing season progresses. An insect herbivore that either reduces or increases competitive asymmetry at a critical stage in the growing season could continue to influence plant competition even after it has stopped feeding. The repeated harvests design does not allow us to fully disentangle the continuous herbivory effect from the effect of harvesting, as it is possible that the lasting effect of herbivory is a consequence of, for example, contrasting regrowth potential in the respective plant species. The consecutive harvests are, however, appropriate when considering the management of the semi-natural grasslands where these species commonly occur in Northern Europe. Interestingly there was an interactive effect between aboveground herbivory and the experimental harvesting of biomass. *Dactylis glomerata* was disfavoured by cutting in control pots, but was unaffected under consecutive harvests when aboveground herbivores were present. For *F. rubra*, however, the trend was the opposite. This demonstrates that insect herbivores may play a balancing role for competitive interactions in managed grasslands that are continuously subjected to an indiscriminate removal of biomass, for instance through cutting. However, a better understanding of the continuous effect of herbivory on competition between the two species would require a design with destructive subsampling of experimental pots.

The benefits and limitations of replacement designs as a means for studying interspecific competition have been discussed extensively (*Cousens, 1996*; *Gibson et al., 1999*; *Jolliffe, 2000*). The purpose of our study was to examine relative differences in competitive outcomes between two plant species under different herbivory and nutrient treatments. Related questions have previously been successfully addressed using a replacement design (*Clay, Marks & Cheplick, 1993*; *Scheublin, Van Logtestijn & Van der Heijden, 2007*; *Garbuzov, Reidinger & Hartley, 2011*; *Sabais et al., 2012*; *Padilla et al., 2013*). A multitude of different indices have been developed for measuring competition between two or more species (reviewed in *Weigelt & Jolliffe, 2003*). When planning a competition experiment, it is therefore important to decide early on, before data collection, which indices may be appropriate for the questions you want to address. We designed and carried out the experiment with the specific intent to use the aggressivity index, as it is a useful index for determining the relative performances of two species growing together (see e.g., *Scheublin, Van Logtestijn & Van der Heijden, 2007*; *Malinowski, Butler & Belesky, 2011*; *Birhane et al., 2014*, for similar usage), and for providing an indicator of competitive asymmetry.

Although nitrogen deposition has previously been shown to cause shifts in both competitive and trophic relationships (*Tylianakis et al., 2008*; *Hoover et al., 2012*), we found no interaction between herbivory and nitrogen availability. Adding nitrogen to experimental mesocosms of marine kelp forests can cause grazers to increase their feeding on weedy species that benefit from nitrogen deposition, thus preventing competitive exclusion of the kelp at high nitrogen levels (*Ghedini, Russell & Connell, 2015*). As we found no effects analogous to this, it is possible that such interactions between nitrogen and herbivory appear only at rates of deposition higher than the ones we simulated. Future assessments of nitrogen deposition effects on competition should aim to investigate larger contrasts in nitrogen levels than we tested here, as we show that a small, but realistic, nitrogen addition appears not to alter the competitive relationship between the two plant species, nor the effect of above- and belowground herbivory on competition. It is also possible that nitrogen effects on biomass production occurred below ground without feeding back into aboveground biomass production for either species. Since our study only includes aboveground biomass responses to herbivory, we cannot elaborate on possible belowground responses of either species. An important next step in studies of plant competition is to develop methods that enable researchers to readily conduct assessments of interspecific competition belowground that are comparable to the standard assessments of interspecific competition aboveground.

## CONCLUSION

We demonstrate a strong increase in aboveground competitive asymmetry between plants exposed to aboveground insect herbivory. As the effect remained after herbivore removal, it appears that insect herbivory can give a plant species a persistent competitive advantage during its establishment. Belowground herbivory appears to counteract the increased competitive asymmetry created by herbivory aboveground. Our results add understanding to, and provide support for, a mechanism underpinning observed shifts in complex plant communities where insect herbivory was experimentally manipulated. We highlight that accurate assessments of the influence of herbivory on plant communities will be achieved only if above- and belowground herbivory effects are investigated jointly.

## ACKNOWLEDGEMENTS

Terhi Korpi, Martin Breunig, Carol Högfeldt and Gerard Malsher assisted in carrying out the experiment. Peter Anderson provided specimens of *S. littoralis*. We are grateful to Vesna Gagic and Mikael Andersson Franko for help with the data analysis, and to Johan Ehrlén, Riikka Kartinen and Gerard Malsher for commenting on the manuscript.

### Funding

Funding was provided by the Swedish research council FORMAS (www.formas.se) to R.B. The funders had no role in study design, data collection and analysis, decision to publish, or preparation of the manuscript.

## Grant Disclosures

The following grant information was disclosed by the authors:
FORMAS.

## Competing Interests

The authors declare there are no competing interests.

## Author Contributions

- Pernilla Borgström conceived and designed the experiments, performed the experiments, analyzed the data, wrote the paper, prepared figures and/or tables, reviewed drafts of the paper.
- Joachim Strengbom and Maria Viketoft conceived and designed the experiments, wrote the paper, reviewed drafts of the paper.
- Riccardo Bommarco conceived and designed the experiments, contributed reagents/materials/analysis tools, wrote the paper, reviewed drafts of the paper.

## Data Availability

  The raw data has been supplied as Data S1.

## Supplemental Information

Supplemental information for this article can be found online at http://dx.doi.org/10.7717/peerj.1867#supplemental-information.

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
