# Peer review of "Aboveground insect herbivory increases plant competitive asymmetry, while belowground herbivory mitigates the effect"

_PeerJ, doi:10.7717/peerj.1867_

## Round 0.1 · original submission · Major Revisions

· Academic Editor

Major Revisions

Both reviewers recognize that the experiments as conducted were robust and that the conclusions drawn flow well from the methods and results.

Reviewer #2 suggests a number of additional experiments (and/or modifications of the described experiments) that would help to provide more information and that would help to clear up some obvious questions.

I do understand that such an undertaking may not be easily possible. However, I would suggest some combination of the following:

1. If the authors had – or have since – collected data in a manner similar to the suggested revisions proposed by Reviewer #2, those data should be included in a revised MS.

2. Can the authors do a better job of pointing to previous studies that have used the same methods?

3. Might the authors recognize the limitations, pointed out by reviewer #2 in their review, and ensure that they are delineated in the MS?

All-in-all, this is a well-written MS, and I don't believe that the the authors overreach in terms of their conclusions (Reviewer #2 even notes that the somewhat tentative tone in parts of the abstract could be reduced in a number of instances). At PeerJ we "...judge the merits of a manuscript based solely on the soundness of its methodology and conclusions...", to quote from the PeerJ FAQ. Both reviewers support the soundness of the existing methodology and the validity of the conclusions. But the suggestions made by Reviewer #2 indicate that further attention is needed to some details.

·

Basic reporting

In “Aboveground insect herbivory increases plant competitive asymmetry, while belowground herbivory mitigates the effect”, Borgstrom et al test whether competitive symmetry between co-occurring species is affected by above ground and belowground herbivory and by nitrogen content in the soil. This is an interesting, well-written and well-explained manuscript.

Experimental design

The hypothesis and predictions are clearly stated and the predictions are explained and supported by previously conducted studies. The methods are clearly described.

Validity of the findings

The statistics are sound for the study.
I appreciated that the authors explained why they used the measure of competition that they did (i.e. the aggressivity index) and that they gave support for using this measure in their study.
One minor issue is that there are no table and figure headings for the supplementary material.

Reviewer 2 ·

Basic reporting

.

Experimental design

.

Validity of the findings

.

Additional comments

In the submitted manuscript by Borgström et al. entitled “Aboveground insect herbivory increases plant competitive asymmetry, while belowground herbivory mitigates the effects” the authors show explore biotic factors in the form of both above- and belowground herbivores and an abiotic factor, soil nitrogen affect competition between two species of grasses. A generalist herbivore, Spodoptera litorallis was used as the aboveground herbivore and wireworm larvae of Agriotes spp. were used as the belowground herbivore. The grasses that were used in the study are Dactylis glomerata and Festuca rubra. The authors did not find any affect of soil nitrogen on competition however they did find that aboveground herbivory caused a 46% decrease in the competitive ability of a grass, F. rubra and a 23% increase in the competitive ability of D. glomerata. They did not find any affect of belowground herbivory, however they did discover that combined above-and belowground herbivory was weaker than predicted. Overall, the authors did a good job of answering the questions that they set out to answer. The experimental design was well planned and robust. However, I do feel that the manuscript could be strengthened with the addition of a few more experiments.

Major Comments
1. The nitrogen treatment used did not alter competition even though previous studies/reports indicate that D. glomerata grows better in nutrient rich soils. This suggests that the Nitrogen treatment that was used may not have been sufficient even though it represents the middle level of nitrogen deposition for West European grasslands. The work will benefit by using a nitrogen treatment level that would allow the authors to see an effect on competition that would correlate with the fact that D. glomerata performs better under nutrient rich conditions. Although the authors do mention this the discussion (Line 343), I feel that it would be appropriate to include an additional level of nitrogen to the experiment.
2. The aboveground herbivore is removed after first harvest and the plants allowed to grow. The belowground herbivores on the other hand were not removed (Line 174 – 176) fearing that their removal will cause damage to the root tissue. However, an easy fix around this would be treating the plants/soil with a contact or systemic insecticide that would kill the belowground herbivore. It would achieve the desired outcome of removal of the belowground herbivore. The authors should incorporate this experiment into the study. This would allow them to investigate whether belowground herbivory affects competitive asymmetry.
3. In the discussion the authors state that F. rubra was greatly preferred by the aboveground herbivores (Line 275 – 276). Presumably this is based on data presented in Fig. 1 and Table 1, which show that aboveground herbivory causes a significantly larger reduction in RY. However, the authors should include a choice test in which the adults are given a choice of feeding between the two species of grasses and determine which grass the S. littoralis larvae prefer. This information will have a major implication on the interpretation of the results. If S. littoralis preferentially feed on F. rubra then the observed competitive asymmetry could be a direct result of host preference by the herbivore. Therefore, I think a choice test showing the preference or non-preference of the grass to the aboveground herbivore should be performed.
4. In calculating the biomass, the grasses were cut 2 cm above the soil surface and dried and weighed. However, the biomass of the roots and the rate of root growth have not been measured. Taking into account only aboveground biomass as measure of performance in response to nitrogen and either herbivory does not provide complete picture on the effect each is having on total plant growth. Given the nature of the experimental design, it is not possible to do so using the same set of plants, however the biomass in the roots in response to either above- or belowground herbivory will be affected and should be measured in a separate set of plants in an independent experiment.

Minor points
1. Line 14 – 20 – The tone in the abstract is very tentative and speculative when in fact many of these facts are established either in the experiments in the paper or from previous studies. I suggest that the authors revise the sentences to show that. For example line 14-20 can be rewritten as “The effect of herbivory on competition likely depends on whether the plants are subjected to aboveground or belowground herbivory or both, and also depends on soil nitrogen levels. It is unclear how these biotic and abiotic factors interactively affect competition. In a greenhouse experiment, we measured competition between two coexisting grass species that respond differently to nitrogen deposition: Dactylis glomerata L., is competitively favoured by nitrogen addition, and Festuca rubra L., is competitively favoured on nitrogen-poor soils.”
2. The nitrogen treatment used did not alter competition even though previous studies/reports indicate that D. glomerata grows better in nutrient rich soils. This suggests that the Nitrogen treatment that was used may not have been sufficient even though it represents the middle level of nitrogen deposition for West European grasslands. The work would be benefitted by using a nitrogen treatment level that would allow the authors to see an effect on competition that would correlate with the fact that D. glomerata performs better under nutrient rich conditions.
3. The aboveground herbivore is removed after first harvest and the plants allowed to grow. The belowground herbivore on the other hand were not removed (Line 174 – 176) fearing that their removal will cause damage to the root tissue. However, an easy fix around this would be treat the plants/soil with a contact or systemic insecticide that would kill the belowground herbivore. It would achieve the desired outcome of removal of the belowground herbivore. The authors should incorporate this experiment into the study. This would allow them to investigate whether belowground herbivory affects competitive asymmetry.
4. In the discussion the authors state that F. rubra was greatly preferred by the aboveground herbivores (Line 275 – 276). Presumably this is based on data presented in Fig. 1 which shows that aboveground herbivory causes a significantly larger reduction in RY. However, the authors should include a choice test in which the adults are given a choice of feeding between the two species of grasses and determine which grass the S. littoralis larvae prefer. This data could have implications on the interpretation of the experiment and the competitive asymmetry was a result of host preference by the herbivore.
5. Line 118 – Dactylis glomerata can be written as D. glomerata. The full species name is already listed in Line 116.
6. Lines 138 to 144 – These sentences describing the benefits/limitations of using replacements designs should be moved to the discussion. In the Methods section, only the methods need to be listed and not why the method was used.
7. Line 146 – Insert a comma after noctuid moth
8. Line 172 – There is an orphan bracket after 48 hours
9. Line 172 – Insert “were” between individuals and removed.
10. Line 178 – 187 – This is a justification of the method used and also belongs in the discussion and not in the methods section.
11. Line 200 – 205 – Again, these sentences list the justification of the method used. Please move this to discussion.
12. Figure S1 and S2 and tables S3, S4 and S5 need to have titles and legends. I was unable to find these in the documents provided.

---

## Round 0.2 · accepted · Accept

· Academic Editor

Accept

Thank you to the reviewers and to the authors for their careful and comprehensive response. I would ask the authors to please consider releasing the review history of this MS, as it contains information that adds considerable value.

·

Basic reporting

No Comments

Experimental design

No comments

Validity of the findings

In the absence of additional experiments, the authors have sufficiently addressed the concerns raised in the previous review. Appropriate inclusions to the text have been made to the revised manuscript that point out to the strengths and weakness of the experiments. The authors have also sufficiently modified the text to better define the scope of the findings.